# Bioconversion of Fe_3_O_4_ Nanoparticles by Probiotics

**DOI:** 10.3390/ph18040542

**Published:** 2025-04-08

**Authors:** Călina Ciont, Amalia Mesaros, Ana Maria Cocean, Rodica Anita Varvara, Elemer Simon, Lucian Barbu-Tudoran, Florica Ranga, Bernadette-Emoke Teleky, Laura Mitrea, Dan Cristian Vodnar, Oana Lelia Pop

**Affiliations:** 1Department of Food Science, University of Agricultural Sciences and Veterinary Medicine, 400372 Cluj-Napoca, Romania; 2Molecular Nutrition and Proteomics Laboratory, Institute of Life Sciences, University of Agricultural Sciences and Veterinary Medicine, 400372 Cluj-Napoca, Romania; 3Physics and Chemistry Department, Technical University of Cluj-Napoca, 400114 Cluj-Napoca, Romania; 4Electron Microscopy Center, Faculty of Biology and Geology, Babes-Bolyai University, 400006 Cluj-Napoca, Romania

**Keywords:** probiotics, iron oxide nanoparticles, iron bioavailability, absorption, ferric-reducing activity, 4-hydroxyphenylacetic acid

## Abstract

**Background/Objectives**: Iron deficiency anemia remains a primary global health concern, affecting millions worldwide. Despite the widespread availability of iron supplements, their efficacy is often hindered by poor bioavailability and adverse gastrointestinal effects. This study explores the potential of probiotics to enhance the bioavailability of Fe_3_O_4_ NPs through probiotic-mediated mechanisms. **Methods**: *Lactobacillus fermentum*, *Lactobacillus rhamnosus*, and *Lactobacillus plantarum* were utilized to investigate their interactions with Fe_3_O_4_ NPs, synthesized via co-precipitation and characterized using transmission electron microscopy, scanning electron microscopy, Fourier transform infrared spectroscopy, and X-ray diffraction. **Results**: The results indicated that probiotics adhere to the nanoparticle surface, with *L. fermentum* exhibiting the highest adhesion and internalization capacity, leading to a significant increase in 4-hydroxyphenylacetic acid (4-HPLA) production (11.73 ± 0.09 mg/mL at 24 h, *p* < 0.05). Spectroscopic analyses further revealed that probiotic metabolism facilitates the oxidation of Fe_3_O_4_ to Fe_2_O_3_. Additionally, Fe_3_O_4_ nanoparticle-treated cultures demonstrated enhanced bacterial viability and metabolic activity, highlighting a synergistic effect between probiotics and iron nanoparticles. **Conclusions**: These findings provide compelling evidence for probiotic-assisted iron supplementation as a promising strategy to enhance iron bioavailability while mitigating the gastrointestinal side effects of conventional iron supplements.

## 1. Introduction

Iron deficiency is one of the most widespread nutritional deficiencies globally, affecting millions of individuals, particularly including vulnerable populations such as pregnant women, children, and individuals with chronic diseases [1]. The World Health Organization (WHO) estimates that 24.8% of the population (1.62 billion people) suffers from anemia, a condition primarily caused by iron deficiency [2]. As iron is fundamental in oxygen transport, redox reactions, energy metabolism, and enzymatic functions, iron deficiency anemia is quite detrimental and develops when iron levels are insufficient, leading to fatigue, weakness, cognitive impairment, and increased susceptibility to infections [3].

As such, various supplementation strategies have been developed to address iron deficiency, including ferrous sulfate, iron salts, and iron oxide nanoparticles (Fe_3_O_4_ NPs) [4]. Although existing conventional iron supplements have been effective in restoring iron levels, they often come with significant drawbacks, such as gastrointestinal discomfort, low bioavailability, and interactions with dietary components. As a result, Fe_3_O_4_ NPs have been acknowledged as an innovative alternative due to their stability, biocompatibility, and ability to resist degradation in the stomach’s acidic environment [5], offering Fe_3_O_4_ NPs promising advantages. However, their absorption in the gastrointestinal tract remains a significant challenge, requiring innovative strategies to enhance their bioavailability. One such approach involves using acid-resistant probiotics that can survive gastric transit and maintain functional viability in the gut, thereby supporting nutrient absorption. *Lactobacillus rhamnosus*, particularly strain GG, is well known for its acid tolerance and ability to adhere to intestinal epithelial cells. These properties support its survival in the gastrointestinal environment and its efficacy in promoting intestinal health and improving the absorption of micronutrients such as iron [6,7]. *Lactobacillus fermentum* and *Lactobacillus plantarum* have also shown excellent survival in simulated gastric juice, with viability rates reaching up to 86.7% and 96.97%, respectively [8,9,10].

Emerging research suggests that probiotics can significantly enhance iron bioavailability by improving solubilization, facilitating absorption, and reducing iron-related oxidative stress [11,12,13,14,15]. Certain probiotic strains, such as *L. fermentum*, *L. acidophilus*, and *Bifidobacterium breve*, have demonstrated the ability to modulate iron metabolism through multiple mechanisms [11]. These bacteria lower the intestinal pH, primarily through acid production, which enhances iron solubility and promotes gut absorption. One of their key excreted metabolites, *p*-hydroxyphenyllactic acid (HPLA), produced by *L. fermentum*, exhibits potent ferric-reducing activity under acidic conditions. Studies have shown that HPLA efficiently reduces ferric iron (Fe^3+^) to its more bioavailable ferrous form (Fe^2+^) at pH 3.8, though this activity significantly diminishes at higher pH values [16]. Moreover, probiotics have been found to upregulate divalent metal transporter 1 (DMT1), a key protein responsible for iron uptake in intestinal cells, further improving iron assimilation [3]. In addition to their direct role in iron metabolism, probiotics support the balance of the gut microbiota, which is essential for optimal nutrient absorption [17,18,19]. Reducing intestinal inflammation and preventing microbial imbalances creates a more favorable environment for iron uptake [18]. Integrating probiotics into iron supplementation strategies may enhance iron absorption while minimizing these adverse effects [18,20].

Recent advances in probiotic research have contributed to a deeper understanding of the molecular pathways involved in the probiotic-mediated regulation of iron metabolism [21]. Beyond increasing solubility and reducing ferric iron to its more bioavailable ferrous form, specific probiotic strains have been shown to modulate the expression of host genes involved in iron transport and storage [18,22]. For instance, the upregulation of divalent metal transporter 1 (DMT1) and ferroportin and the modulation of ferritin expression are crucial in enhancing cellular iron uptake and systemic distribution [23]. Moreover, probiotics have demonstrated the ability to influence signaling pathways such as TLR2/NF-κB, involved in immune response and inflammatory regulation, thereby indirectly supporting a gut environment that is more conducive to iron absorption [3,11,13]. In addition, probiotics contribute to intestinal barrier integrity by modulating mucin expression and tight junction proteins such as MUC2, CLDN, and ZO-1, facilitating nutrient permeability and reducing inflammation [17]. Their antioxidant properties also play a role in mitigating the oxidative stress associated with iron metabolism [24], with emerging evidence suggesting that certain lactic acid bacteria may utilize extracellular electron transfer mechanisms to enhance ferric iron reduction, representing a novel metabolic pathway that supports iron solubilization and absorption [13].

The combined use of Fe_3_O_4_ NPs and probiotics as a novel supplementation strategy is gaining increasing attention due to the challenges related to the bioavailability of nanoparticles and the beneficial effects of probiotics on iron absorption. Recent studies have demonstrated that certain probiotic strains, such as *L. plantarum*, enhance iron absorption through multiple mechanisms [22,24]. Specifically, probiotics can regulate the expression and activity of divalent metal transporter 1 (DMT1), an activator protein responsible for iron absorption in the gut, thereby facilitating the absorption of dietary iron [18]. Additionally, probiotics exhibit anti-inflammatory properties that may reduce intestinal inflammation, creating a more favorable environment for optimal nutrient absorption [25]. Studies indicate that probiotics may act as protective carriers for Fe_3_O_4_ NPs, enhancing their stability and absorption in the gastrointestinal tract [26,27]. For example, encapsulating iron dextran with probiotic bacteria in pH-triggered hydrogels prevents iron aggregation in the acidic gastric environment, thereby ensuring efficient delivery to the gut, where absorption occurs [28]. In addition, probiotics may alleviate the oxidative stress and inflammation associated with Fe_3_O_4_ NP administration, supporting their role in promoting iron homeostasis [29]. Despite the present research in this field, a pressing need remains to explore how the probiotic-mediated biotransformation of Fe_3_O_4_ NPs enhances iron absorption.

As such, this study aims to investigate the underlying molecular and biochemical mechanisms by which select probiotic strains, *L. fermentum*, *L. rhamnosus*, and *L. plantarum*, synergistically interact with Fe_3_O_4_ NPs to enhance iron solubilization.

## 2. Results and Discussion

### 2.1. Morphological and Structural Characterization of Fe_3_O_4_ NPs

The physical characteristics of Fe_3_O_4_ NPs, such as their aggregation state and surface area, directly affect their dissolution behavior, redox activity, and interaction with probiotic metabolic pathways [5,30]. Therefore, a detailed morphological and structural analysis was used to establish the Fe_3_O_4_ NPs’ properties after synthesis and their potential bioavailability enhancement via probiotics.

The TEM image (Figure 1A) highlights the nanoscale dimensions of the Fe_3_O_4_ particles, exhibiting a quasi-spherical morphology with an aggregation tendency. This fact can be attributed to a direct consequence of strong dipole–dipole magnetic interactions, which is a common feature of superparamagnetic iron oxide nanoparticles [31]. The uniform particle distribution suggests a controlled synthesis process. The SEM image (Figure 1B) provides a detailed view of the surface topology, emphasizing the rough and clustered structure of the Fe_3_O_4_ NPs. The degree of aggregation observed in the SEM image suggests a high surface-area-to-volume ratio, which can benefit probiotic adhesion and biofilm formation [32]. The clustered porous nature of Fe_3_O_4_ NPs provides a favorable environment for probiotic adhesion, where bacterial exopolysaccharides and surface proteins can establish strong interactions with the nanoparticles, facilitating biofilm-mediated iron uptake. This is consistent with studies that have shown probiotics to be able to significantly enhance intestinal iron absorption by releasing metabolites that facilitate this conversion [3,28]. Moreover, the degree of Fe_3_O_4_ NP aggregation influences the NPs’ behavior in simulated gastrointestinal conditions, where probiotic-secreted metabolites may act as natural dispersing agents, preventing excessive nanoparticle clustering and ensuring sustained iron release [33].

Furthermore, the complementary structural and chemical data of the synthesized Fe_3_O_4_ NPs were investigated through FT-IR and XRD methods. The FT-IR spectrum (Figure 2A) displays an absorption band near 570 cm^−1^, which is typically assigned to the Fe–O stretching vibration in magnetite, confirming the formation of the Fe_3_O_4_ phase. Subsequent FT-IR analyses confirmed that no residual ethanol or ethyl acetate reagents were detectable on the nanoparticle surfaces. Meanwhile, the XRD pattern (Figure 2B) confirmed the inverse spinel crystal structure characteristic of magnetite, indicated by the intense diffraction peaks at 2θ values corresponding to planes such as (220), (311), (400), (422), (511), and (440). The sharpness and intensity of these peaks attest to the high crystallinity of the NPs. At the same time, the average crystallite size, determined by the Debye–Scherrer formula (16.9 nm), places them in the nanometer range.

Therefore, the synergistic results collected from the morphological and structural characterization of the Fe_3_O_4_ NPs underscore the successful synthesis of magnetite nanoparticles via co-precipitation.

### 2.2. Impact of Fe_3_O_4_ NPs on Bacterial Fermentation

The presence of Fe_3_O_4_ NPs in fermentation systems has been explored by examining how these nanoparticles influence bacterial physiology, fermentation kinetics, and the ultimate bioavailability of iron in the context of probiotic-mediated iron supplementation. Iron is a vital cofactor for numerous enzymes that drive energy production, growth, and overall fermentative performance in probiotic bacteria [4,29,34]. By modulating iron availability, Fe_3_O_4_ NPs may stimulate key metabolic pathways, potentially boosting the fermentation rate and efficiency [16,22,30].

SEM images (Figure 3) confirmed the adhesion of Fe_3_O_4_ NPs to the surfaces of all three probiotic strains, with *L. fermentum* exhibiting the most pronounced attachment. Variations in cell surface charge, hydrophobicity, or extracellular polymeric substances may influence this strain-dependent affinity [35]. This interaction likely stems from the binding affinities between the nanoparticle surface functional groups (hydroxyls) and the exopolysaccharides or proteinaceous layers on the bacterial cell wall [36]. Similar adherence has been suggested to facilitate localized iron reduction and uptake, as probiotic cells secrete organic acids and siderophores that enhance iron solubility [35,36].

The TEM results (Figure 4) further revealed the internalization of Fe_3_O_4_ NPs within *L. fermentum* cells, while *L. plantarum* and *L. rhamnosus* showed lower degrees of internalization. Notably, *L. fermentum* maintained a smooth cell surface post-exposure, indicating potential membrane adaptations that facilitate nanoparticle uptake without compromising cellular integrity. These findings align with those of previous researchers investigating maghemite (γ-Fe_2_O_3_) nanoparticles in combination with *L. fermentum*. They found that these nanoparticles not only adhered to the bacterial surface but also significantly improved the accumulation of magnetic particles in the gastrointestinal tract, supporting their potential as iron delivery systems [4,37].

In contrast, *L. rhamnosus* exhibited signs of osmotic stress upon exposure to Fe_3_O_4_ NPs. SEM images revealed morphological changes indicative of cell stress, corroborated by a TEM analysis showing an increased presence of ribozymes, thus suggesting upregulated stress-related metabolic activity. These observations imply that Fe_3_O_4_ NPs may impose a metabolic burden on *L. rhamnosus*, potentially affecting its viability and probiotic functionality. For instance, it has been shown that *L. rhamnosus* can regulate mRNA abundance to recover translation efficiency under osmotic stress, an essential parameter for maintaining cellular functions during stress conditions [38,39]. The coordinated increase in transcript abundance ensures that ribosomes have a readily available pool of mRNAs encoding proteins that are critical for stabilizing the cell membrane, preventing protein aggregation, and maintaining overall cellular homeostasis [40]. Fe_3_O_4_ NPs have also been described to adhere to the surface of probiotics, thus enhancing their solubility and bioavailability [41,42]. *Lactobacillus* and *Bifidobacterium* species have been stated to possess surface structures (exopolysaccharides, lipoteichoic acids, surface proteins) that enable adhesion to mineral surfaces [42]. Nevertheless, beyond these physicochemical properties, strain-specific differences in gene expression and metabolic pathways may explain these effects [43,44]. For example, *Lactobacillus* spp. may upregulate genes (expression of *lar*) involved in organic acid production [44], thereby enhancing Fe_3_O_4_ solubility, while *Bifidobacterium* spp. might rely on iron-binding proteins for nanoparticle interaction [43]. Another key factor could be the differential expression of siderophore-related genes, as most probiotic bacteria have been found to lack traditional siderophores [45,46]. However, the *Lactobacillus* genus has been reported to produce siderophores (peptides) that can chelate Fe^3+^ and facilitate transport [46]. Additionally, gene expression in metal ion homeostasis, such as that relating to DNA-binding proteins from starved cells and ferritin-like proteins, may contribute to intracellular iron sequestration and storage [46,47].

The presence of Fe_3_O_4_ NPs notably influenced the fermentation profiles of the tested probiotic strains, suggesting a potential synergistic relationship between iron oxide nanostructures and probiotic metabolism. As shown in Figure 5, each bacterial species (*L. plantarum*, *L. rhamnosus*, and *L. fermentum*) exhibited distinct viability patterns in response to Fe_3_O_4_ supplementation over the incubation period. Fe_3_O_4_ NPs influenced the cell viability relative to that of the control (bacteria without nanoparticles), particularly at the end of fermentation. As incubation progressed, *L. plantarum* consistently exhibited higher viable cell counts when Fe_3_O_4_ NPs were added, as compared to the control. Similar trends were observed for *L. fermentum* viability. This enhancement became more pronounced at the end time points (12–24 h), suggesting that the nanoparticle-provided iron may have bolstered key metabolic pathways involved in cell division and energy production [48]. Nonetheless, the results showed that iron supplementation via nanoparticles did not inhibit bacterial viability. On the other hand, the elevated viable cell counts in Fe_3_O_4_-supplemented cultures at the mid-to-late fermentation stages suggest that *L. rhamnosus* can harness the iron from the nanoparticles to sustain growth. Moreover, differences among the three species’ response profiles highlight strain-specific iron acquisition and regulation pathways. This observation resonates with the concept that certain probiotics possess specialized mechanisms (iron chelation or reduction) to facilitate iron acquisition from mineral complexes.

Regarding pH levels during the cultivation of the bacteria with or without Fe_3_O_4_ NPs, it was noticeable that the three bacteria had similar behaviors regarding their acidification profiles. A more rapid decline in pH values occurred when Fe_3_O_4_ NPs were present, although the final pH values remained similar. Even though the cultures of *L. plantarum* and *L. rhamnosus* with Fe_3_O_4_ showed a more rapid pH drop, the bacterial proliferation was lower than that of cultures without Fe_3_O_4_, which suggests that the Fe_3_O_4_ NPs may have slowed down their proliferation. Even more interestingly, the bacterial proliferation of *L. fermentum* with Fe_3_O_4_ was slightly higher than that of *L. fermentum* without Fe_3_O_4_, and the pH values followed the same trend as in the cases of the other two bacteria. These differences likely reflect inherent species-level variations in carbohydrate metabolism, acid production, and iron uptake by bacteria [4]. Furthermore, the notable drop in pH aligns with the enhanced growth observed in the Fe_3_O_4_-supplemented cultures by the end of the fermentation, supporting the conclusion that nanoparticle-provided iron catalyzes more vigorous fermentation.

### 2.3. Ferric-Reducing Activity of Probiotics

The ferric-reducing power of certain probiotic bacteria is central to improving iron absorption in the gut. In simple terms, most dietary iron is ingested as ferric iron (Fe^3+^), which is not easily absorbed by the body. For iron to be taken up by intestinal cells, it must first be reduced to its ferrous form (Fe^2+^). This conversion is crucial because Fe^2+^ can be readily transported into enterocytes via DMT1. In this respect, it was shown that the reduction efficiency was mainly influenced by the probiotic production of metabolites like 4-HPLA and the pH [16].

The metabolism of three probiotic strains, *L. plantarum*, *L. rhamnosus*, and *L. fermentum*, was analyzed under fermentation conditions in the presence of Fe_3_O_4_ NPs (Table 1). Several parameters were monitored, including energy substrate utilization (glucose and fructose), the production of organic acids (lactic, acetic, and propionic acids), the accumulation of 4-HPLA, and ethanol formation.

The observed progressive decline in glucose and fructose levels indicates active bacterial metabolism, consistent with the fermentative pathways characteristic of lactic acid bacteria. *L. rhamnosus* and *L. fermentum* initially had higher sugar concentrations than *L. plantarum*. However, after 24 h, the sugar dropped to low levels, indicating complete fermentation. Notably, the incorporation of Fe_3_O_4_ NPs did not restrict sugar metabolism, whereas, for *L. rhamnosus*, a slight enhancement in the rate of sugar consumption was observed. These findings align with those of studies suggesting that metal-based nanoparticles can modulate bacterial metabolism [34,49]. The impact of Fe_3_O_4_ NPs on bacterial activity is highly dependent on their physicochemical properties, particularly their particle size, surface charge, and synthesis method [50]. Shen et al. (2018) [51] showed that Fe_3_O_4_ NPs synthesized by co-precipitation exhibit high aqueous medium dispersions, which can facilitate their interaction with microbial cells, potentially modulating metabolic pathways [51].

One of the most significant metabolites excreted after the fermentation was 4-HPLA, detected in the highest concentration for *L. fermentum*, significantly exceeding the levels observed for *L. plantarum* and *L. rhamnosus*. Previous studies have suggested that 4-HPLA significantly affects iron metabolism, particularly concerning ferric reduction [4,16,23]. This metabolite has been reported to efficiently reduce Fe^3+^ to Fe^2+^ under acidic conditions, potentially enhancing iron’s bioavailability in the gut. González et al. (2017) [16] investigated the iron-reducing capacity of *L. fermentum* CECT5716, indicating that this strain excretes 4-HPLA in significant amounts, mimicking the function of duodenal cytochrome B (DcytB), an endogenous iron-reducing enzyme in enterocytes. Their findings confirmed that 4-HPLA actively reduces iron forms at pH 3.8, making iron more bioavailable for absorption. Moreover, they demonstrated that the iron-reducing activity of *L. fermentum* is predominantly extracellular, as evidenced by the retained reducing power in the cell-free supernatant [16]. This aligns with our results, where *L. fermentum* indicates the highest levels of 4-HPLA, further reinforcing its role in iron metabolism.

Beyond its role in iron reduction, 4-HPLA has also been associated with modulating organic acid metabolism. Previous studies have shown that elevated 4-HPLA concentrations correlate with increased acetate accumulation, suggesting that this metabolite influences key pathways involved in organic acid biosynthesis [52]. Acetate is a key metabolic byproduct in lactic acid bacterium fermentation and serves as an essential precursor for energy metabolism and bacterial growth regulation [3,23,24]. Higher acetate levels contribute to a more favorable intestinal environment by modulating the pH and supporting the growth of beneficial microbiota [30]. Moreover, probiotic fermentation contributes to gut acidification by producing organic acids (short-chain fatty acids), which enhance iron solubility and absorption [53]. The reduction in pH due to short-chain fatty acid production has been suggested to directly influence iron solubility in the intestinal environment. For instance, lower fecal pH levels, which have been linked to probiotic administration, enhance iron absorption by mechanisms that facilitate the dissolution of ferric iron into its soluble form [54]. Ji et al. (2021) [55] also established that elevated levels of *Corynebacterium* and *Lactobacillus* (5 × 10^9^) species enhance the production of short-chain fatty acids, which subsequently stimulate interferon-β secretion in alveolar macrophages, thereby influencing iron absorption. Additionally, probiotic immunomodulation exerts anti-inflammatory effects, which downregulate hepcidin. This relationship was further supported by Vonderheid et al. (2019) [56] and Rusu et al. (2020) [23], who independently confirmed the positive correlation between probiotic supplementation and improved iron uptake. Specifically, *L. plantarum* 299v (10^9^ CFU/g) has been shown to lead to a significant increase in ferritin levels in women of reproductive age (mean increase 2.45 ng/mL, 95% CI 0.61–4.3, n = 320) [24]. Furthermore, *L. curvatus* (10^9^ CFU/g) with Fe gluconate at 120 mg Fe/kg diet supplementation re-established anemia parameters in 8 weeks of oral treatment [57]. These mechanisms collectively contribute to improved iron absorption and probiotic viability, resulting in a synergistic effect. Thus, probiotics influence iron metabolism through multiple mechanisms, including pH reduction, organic acid secretion, and direct iron chelation via metabolites like 4-HPLA [4,16,22,30]. These combined effects create an environment conducive to increased iron bioavailability, highlighting the role of *L. fermentum* in iron homeostasis [5].

FT-IR and XRD analyses were conducted to confirm the structural and chemical transformation of Fe_3_O_4_ NPs into Fe_2_O_3_ during probiotic fermentation. The FT-IR spectrum (Figure 6A) provides the results of this structural transition. The characteristic Fe–O stretching vibration typically observed around 570 cm^−1^ in pure magnetite was slightly shifted across the different bacterial treatments, with the most pronounced alteration occurring in *L. fermentum*-treated samples. This indicates structural perturbations resulting from bacterial enzymatic or metabolic activity, particularly the reduction of Fe^3+^ to Fe^2+^. Additionally, variations in absorbance intensity suggest differential interactions at the bacterial cell–nanoparticle interface, likely mediated by secreted metabolites, siderophores, or reductases. Previous studies have demonstrated that bacterial strains with high ferric reductase activity can modulate the electronic structure of iron oxides, thereby enhancing their dissolution [12,48]. Beyond the Fe–O vibrational region, other subtle spectral changes can be noted in the 1200–1600 cm^−1^ range, where bacterial exopolysaccharides, proteins, and other organic compounds exhibit characteristic absorbance peaks. These features are particularly relevant because bacterial surface biomolecules can act as ligands that chelate iron, further enhancing its solubility and bioavailability. The observed spectral changes suggest the involvement of multiple biogenic factors in iron reduction.

The XRD pattern further corroborated the microbial-induced oxidation process. The hexagonal hematite structure of Fe_2_O_3_ (α-Fe_2_O_3_) formed after fermentation, and the crystalline phase was analyzed (Figure 6B). The reflections were assigned to Miller indices (104), (110), (113), (024), (116), (018), (214), (030), which could be indexed by the JCPDS sheet—PDF no. 20-1096. The results suggest substantial oxidation of Fe_3_O_4_ (magnetite) into Fe_2_O_3_ (hematite) in the case of *L. fermentum* and *L. plantarum* fermentation. The disappearance of magnetite-specific peaks and the emergence of hematite-specific diffraction patterns indicate a complete phase transformation. This transformation is driven by the acidic and redox-active microenvironment generated by a high concentration of 4-HPLA. The increased crystallinity of *L. fermentum*–Fe_2_O_3_ post-fermentation suggests a controlled oxidation mechanism, likely involving direct microbial interactions with the nanoparticle surface. From a mechanistic perspective, the microbial secretion of metabolites capable of chelating Fe^3+^ plays a crucial role in this transformation, with *L. fermentum* known to enhance iron bioavailability through extracellular electron transfer [13,16]. On the other hand, *L. rhamnosus* and *L. plantarum* induce noticeable modifications and exhibit less pronounced peak broadening than *L. fermentum*, suggesting a more moderate interaction with the iron oxide lattice. These results suggest that probiotic-fermented Fe_3_O_4_ NPs could be an effective dietary iron source.

These study findings should be interpreted with consideration of their limitations. First, although 4-HPLA has been identified as a key metabolite in iron reduction, a direct functional validation of its role in promoting iron absorption was not conducted herein. As such, future in vitro assays using intestinal cell models must confirm its biological activity. Second, additional in vivo studies are necessary to assess the impact of probiotic-fermented Fe_3_O_4_ NPs on iron bioavailability. Third, the results were obtained using specific probiotic strains (*L. fermentum*, *L. plantarum*, and *L. rhamnosus*), and further studies should investigate whether strain-specific metabolic variations influence the strains’ ability to enhance iron absorption. While several studies have explored the influence of probiotics on iron bioavailability, less attention has been given to how this intervention may affect other key probiotic functions, such as immunomodulation, enhancement of gut barrier integrity, and modulation of the host microbiota [58,59]. This represents an important gap, and it remains unclear whether co-administration of iron and probiotics may significantly alter the functional properties of probiotic strains, either through direct metabolic interactions or by influencing the intestinal environment. Future studies should aim to elucidate these potential effects in order to ensure that improvements in iron status do not come at the expense of other health-promoting activities of probiotics.

## 3. Materials and Methods

### 3.1. Materials

Iron(II) sulfate heptahydrate, FeSO_4_·7H_2_O (ACS, 98–102%, VWR, Radnor, PA, USA); potassium nitrate, KNO_3_ (Sigma Aldrich, 99%; St. Louis, MO, USA); potassium hydroxide, KOH (Alfa Aesar, 99.8%; Haverhill, MA, USA); and ethanol absolute were used without further purification.

The fermentation process involved three probiotic strains, *Lactobacillus fermentum* (LMG 6902), *Lactobacillus plantarum* (ATCC 8014), and *Lactobacillus rhamnosus* (LMG 25626), acquired from the BCCM/LMG Bacteria Collection. The bacterial cultures were cultivated in de Man, Rogosa, and Sharpe (MRS) broth, supplied in dried form by HIMEDIA (Einhausen, Germany). Acetonitrile, HPLC gradient-grade, was provided by Merck (Darmstadt, Germany), while water was purified with a Direct-Q UV system by Millipore (Burlington, MA, USA). Pure standards of glucose, fructose, lactic acid, acetic acid, propionic acid, 4-hydroxyphenylacetic acid (4-HPLA), and ethanol (>98%) were purchased from Tokyo Chemical Industry (Tokyo, Japan).

### 3.2. Synthesis of Iron Oxide Nanoparticles

The co-precipitation method was applied to obtain Fe_3_O_4_ NPs. Briefly, FeSO_4_·7H_2_O was dissolved in MiliQ water at a molar ratio of 1:1 under continuous stirring to obtain a homogeneous solution [21]. The reaction underlying the formation of the iron oxide matrix (Equation (1)) is based on the partial oxidation of divalent iron to trivalent iron, simultaneously with the reduction of the nitrate ion:3FeSO_4_ + 2KNO_3_ + 4KOH → Fe_3_O_4_↓+ 3K_2_SO_4_ + 2NO_2_↑ + 2H_2_O(1)

The mixture was heated to 80 °C, and 25% NH_3_·H_2_O was added dropwise until the pH reached 11, ensured by the presence of OH^−^ ions. The reaction mixture was maintained in these conditions for 60 min, with constant stirring to promote crystal growth and maturation. The resulting black precipitate was magnetically separated using a 0.3 T magnetic field, then thoroughly washed with MiliQ water and a 4:1 ethanol and ethyl acetate combination (*v*/*v*). The produced Fe_3_O_4_ NPs were suspended in MiliQ water, resulting in a stable black solution.

### 3.3. Characterization of the Fe_3_O_4_ NPs

Transmission electron microscopy (TEM) and scanning electron microscopy (SEM) analyses were conducted using a HITACHI HD-2700 STEM microscope (Hitachi, Tokyo, Japan) equipped with a digital image recording system alongside an SU8230 scanning electron microscope (SEM) from the same manufacturer [19]. Dehydration was achieved using an ethanol series (30%), with SEM samples dried using a liquid CO_2_ critical point dryer (1200 psi, 31 °C, 30 min) to eliminate surface tension artifacts, while TEM samples were air-dried on carbon-coated 400-mesh copper grids under low humidity with 0.1% BSA to prevent aggregation. The electron microscope was coupled with an Aztec X-Max 1160 EDX detector (Oxford Instruments, Abingdon, UK) for energy-dispersive X-ray spectroscopy (EDS). SEM/EDS images were acquired under 30 kV and 10 μA operating conditions. Image acquisition and particle size measurements were performed using Hitachi acquisition software (version 8.1). To ensure statistical reliability, n ≥ 100 measurements were made across multiple images, and the data were fitted to a Gaussian distribution using Origin 2019b (ver. 9.65) [60].

Fourier transform infrared spectroscopy (FT-IR) was conducted to analyze the chemical nature of the nanoparticle surfaces using a Shimadzu IR-Prestige FTIR Spectrometer equipped with a diamond PIKEMIRacle single reflection plate unit. The spectra were taken in the 600–4000 cm^−1^ range with a resolution of 4 cm^−1^ [61].

An X-ray diffraction (XRD) analysis was performed at ambient temperature using a Bruker D8 Advance diffractometer to investigate the crystal structure of the synthesized Fe_3_O_4_ NPs. The instrument was equipped with Cu Kα radiation (λ = 1.5406 Å), and diffraction patterns were recorded over a suitable 2θ range to capture all characteristic peaks. Subsequently, the Scherrer method was used to estimate the average crystallite size [62].

### 3.4. Probiotic Inoculum Preparation with Fe_3_O_4_ NPs

The lactic acid bacterium strains *L. fermentum* LMG 6902, *L. rhamnosus* LMG 25626, and *L. plantarum* ATCC 8014 were cultivated under aerobic conditions in de Man, Rogosa, and Sharpe (MRS) broth medium. The medium was sterilized by autoclaving at 121 °C for 15 min. The bacterial cultures were incubated at 37 ± 1 °C for 24 h, followed by an additional inoculum and incubation period under the same conditions until they reached the stationary growth phase (~20 h post-inoculation) [63]. The bacterial suspensions were assessed for viable cell counts, reaching approximately 10 log CFU/mL. The bacterial optical density (OD) at 600 nm (OD_600_) was measured using a NanoDrop 1000 spectrophotometer (NanoDrop Technologies, Wilmington, DE, USA). The recorded OD_600_ values ranged between 0.009 and 0.011. For Fe_3_O_4_ nanoparticle (NP) exposure, a 100 mL bacterial suspension was supplemented with 120 µg/mL of Fe_3_O_4_ NPs [64]. The bacterial NP suspension was then incubated at 37 ± 1 °C for 24 h under static conditions to allow potential bacterial interactions and uptake of Fe_3_O_4_ NPs. All these processes were carried out in a sterile environment.

### 3.5. Ferric-Reducing Activity of Probiotics

#### 3.5.1. Secondary Metabolite Identification

To identify excreted bacterial compound(s) with ferric-reducing activity, the samples were centrifuged at 12,298× *g* and 4 °C for 30 min [65]. The supernatant was then filtered through a 0.45 μm nylon filter and analyzed using an HPLC system. The analysis was performed using an HP-1200 liquid chromatograph equipped with a quaternary pump, autosampler, DAD detector, and MS-6110 single-quadrupole API–electrospray detector (Agilent Technologies, Clara, CA, USA). The separation of compounds was performed on a Polaris Hi-Plex H column (300 × 7.7 mm, Agilent Technologies), employing a mobile phase of 5 mM H_2_SO_4_ at a rate of 0.6 mL/min. The column temperature was adjusted to 80 °C, while the RID temperature was maintained at 35 °C. The elution process for the compounds took 25 min. Data were acquired and interpreted using the OpenLab software ChemStation (Agilent Technologies, Clara, CA, USA). The identified compounds in the analyzed samples were recognized by comparing the duration of their stay within the system with those of standard reference compounds. The compounds analyzed during fermentation included glucose, fructose, lactic acid, acetic acid, propionic acid, 4-HPLA, and ethanol. Chromatograms were recorded at wavelengths of λ = 210 and 280 nm, and data acquisition was performed with Agilent ChemStation software (ver. C.01.08).

#### 3.5.2. Characterization of the Probiotic Biomass Enriched with Fe_3_O_4_ NPs

The structural, morphological, and chemical properties of Fe_3_O_4_ nanoparticle-enriched probiotic biomass were characterized using electron microscopy, Fourier transform infrared spectroscopy, and X-ray diffraction techniques [66,67].

### 3.6. Statistical Analysis

All the values are displayed as means ± standard deviations (SDs) from three independent experiments. All continuous variables were tested for normality using the Shapiro–Wilk test before the statistical analysis. Data following a normal distribution were analyzed using a one-way analysis of variance (ANOVA), Tukey’s comparison test, and post hoc Dunnett’s multiple comparison tests. Minitab statistical software (version 16.1.0; LEAD Technologies, Inc., Charlotte, NC, USA) and Graph Prism Version 8.0.1. (GraphPad Software Inc., San Diego, CA, USA) were used to analyze the differences among samples with significance levels of *p* < 0.05. Statistical significance was assumed at the 95% confidence level for differences in mean values.

## 4. Conclusions

The present study validates the hypothesis that probiotic intervention significantly enhances the bioavailability of Fe_3_O_4_ NPs through several synergistic mechanisms. Probiotics, particularly *L. fermentum*, exhibit strong adhesion and internalization of the nanoparticles, facilitating more effective interactions at the microbial interface. This strain’s elevated production of 4-HPLA plays a crucial role in reducing ferric ions, thus improving iron solubility and subsequent uptake. Additionally, spectroscopic analyses revealed microbial-induced oxidation of Fe_3_O_4_ to Fe_2_O_3_, which further contributes to enhanced iron release and bioavailability. These findings underscore the potential of integrating probiotic formulations with iron oxide nanoparticles as a novel supplementation strategy to overcome the limitations of traditional iron therapies, mainly by reducing gastrointestinal side effects while optimizing iron absorption. Future research should focus on elucidating the precise molecular pathways involved and validating these results in clinical settings.

## Figures and Tables

**Figure 1 pharmaceuticals-18-00542-f001:**
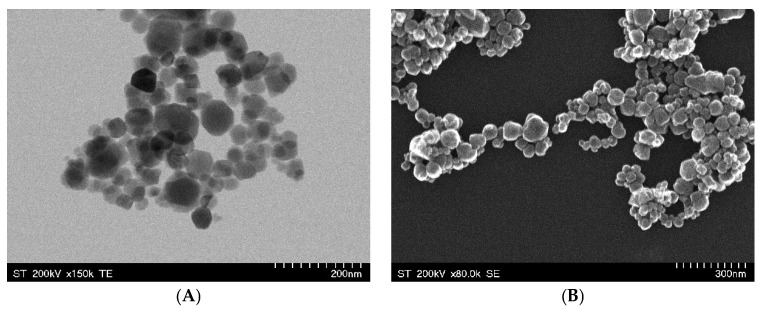
TEM (**A**) and SEM (**B**) images of Fe_3_O_4_ NPs synthesized by the co-precipitation method.

**Figure 2 pharmaceuticals-18-00542-f002:**
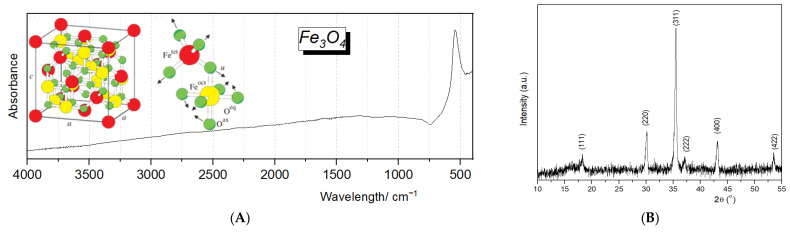
FT-IR spectra (**A**) and XRD patterns (**B**) of Fe_3_O_4_ NPs synthesized by the co-precipitation method.

**Figure 3 pharmaceuticals-18-00542-f003:**
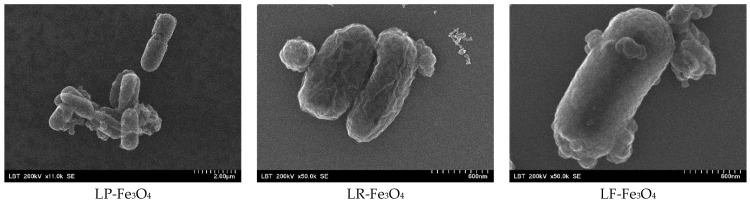
SEM images of probiotics with Fe_3_O_4_ NPs: LP-Fe_3_O_4_ for *L. plantarum* after fermentation with Fe_3_O_4_ NPs; LR-Fe_3_O_4_ for *L. rhamnosus* after fermentation with Fe_3_O_4_ NPs; LF-Fe_3_O_4_ for *L. fermentum* after fermentation with Fe_3_O_4_ NPs.

**Figure 4 pharmaceuticals-18-00542-f004:**
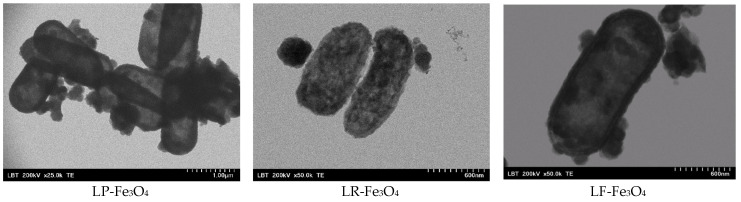
TEM images of probiotics with Fe_3_O_4_ NPs: LP-Fe_3_O_4_ for *L. plantarum* after fermentation with Fe_3_O_4_ NPs; LR-Fe_3_O_4_ for *L. rhamnosus* after fermentation with Fe_3_O_4_ NPs; LF-Fe_3_O_4_ for *L. fermentum* after fermentation with Fe_3_O_4_ NPs.

**Figure 5 pharmaceuticals-18-00542-f005:**
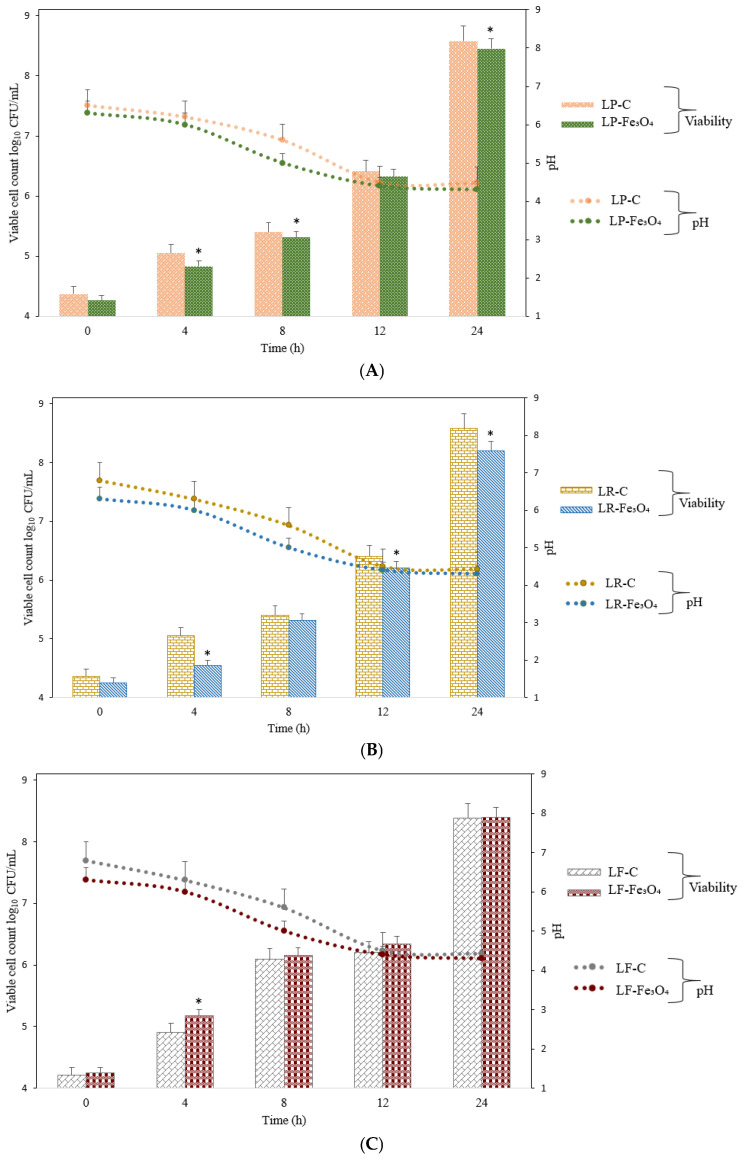
Viable cell counts for *L. plantarum*, *L. rhamnosus*, and *L. fermentum* in the presence of Fe_3_O_4_ NPs. (**A**) *L. plantarum* cultured under control conditions (LP-C) and in the presence of Fe_3_O_4_ NPs (LP-Fe_3_O_4_) for 24 h. (**B**) *L. rhamnosus* cultured under control conditions (LR-C) and in the presence of Fe_3_O_4_ NPs (LR-Fe_3_O_4_) for 24 h; (**C**) *L. fermentum* cultured under control conditions (LF-C) and in the presence of Fe_3_O_4_ NPs (LF-Fe_3_O_4_) for 24 h. The values for bacterial viable cell growth and pH are displayed as mean values ± SDs, log10 CFU/g, n = 3 (CFU/mL—colony-forming units/mL of the sample). A two-way ANOVA was applied to compare differences between the bacterial species through the Tukey multiple comparisons method with a *p*-value provided if *p* < 0.05, and the symbols for the Tukey interpretation are as follows: * *p* < 0.05. LP-C for *L. plantarum* (control); LR-C for *L. rhamnosus* (control); LF-C for *L. fermentum* (control); LP-Fe_3_O_4_ for *L. plantarum* after fermentation with Fe_3_O_4_ NPs; LR-Fe_3_O_4_ for *L. rhamnosus* after fermentation with Fe_3_O_4_ NPs; LF-Fe_3_O_4_ for *L. fermentum* after fermentation with Fe_3_O_4_ NPs.

**Figure 6 pharmaceuticals-18-00542-f006:**
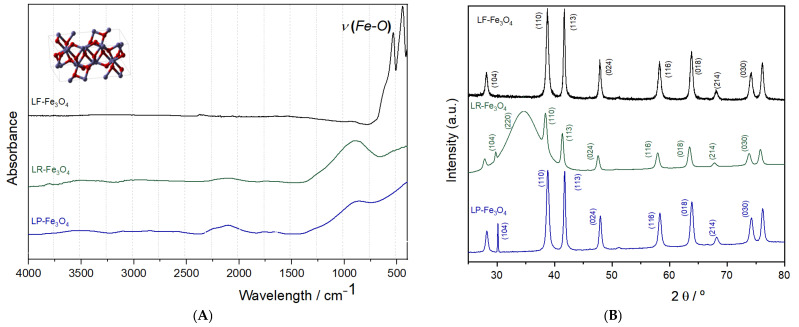
FT-IR spectra (**A**) and XRD patterns (**B**) of Fe_3_O_4_ NPs after fermentation with probiotic bacteria. LP-Fe_3_O_4_ for *L. plantarum* after fermentation with Fe_3_O_4_ NPs; LR-Fe_3_O_4_ for *L. rhamnosus* after fermentation with Fe_3_O_4_ NPs; LF-Fe_3_O_4_ for *L. fermentum* after fermentation with Fe_3_O_4_ NPs.

**Table 1 pharmaceuticals-18-00542-t001:** Secondary metabolite (mg/mL) profiling of *L. plantarum*, *L. rhamnosus*, and *L. fermentum* after fermentation with Fe_3_O_4_ NPs.

Compound	Time	*L. plantarum*		*L. rhamnosus*		** *L. fermentum* **	
		LP-C	LP-Fe_3_O_4_	LR-C	LR-Fe_3_O_4_	LF-C	LF-Fe_3_O_4_
Fructose	0	2.39 ± 0.01	2.28 ± 0.04	4.91 ± 0.03	4.47 ± 0.059	7.74 ± 0.17	7.70 ± 0.15
	4	1.50 ± 0.10	1.56 ± 0.03	3.17 ± 0.10	3.69 ± 0.041	3.17 ± 0.14	3.19 ± 0.10
	8	0.86 ± 0.07	0.90 ± 0.10	2.00 ± 0.05	2.71 ± 0.050	2.33 ± 0.25	2.30 ± 0.05
	12	0.48 ± 0.09	0.34 ± 0.12	1.91 ± 0.02	1.47 * ± 0.074	1.65 ± 0.14	1.60 ± 0.04
	24	0.33 ± 0.08	0.38 ± 0.11	0.63 ± 0.14	N.D.	0.26 ± 0.04	0.28 ± 0.08
Glucose	0	19.54 ± 0.02	19.61 ± 0.14	19.73 ± 0.12	19.549 ± 0.04	19.06 ± 0.09	18.86 ± 0.05
	4	13.71 ± 0.07	13.53 ± 0.25	18.41 ± 0.04	18.607 ± 0.09	16.68 ± 0.14	16.55 ± 0.10
	8	7.65 ± 0.02	7.49 ± 0.11	17.32 ± 0.11	17.395 ± 0.36	5.64 ± 0.10	5.60 ± 0.11
	12	5.88 ± 0.10	5.91 ± 0.11	14.42 ± 0.20	3.047 * ± 0.08	2.52 ± 0.08	2.50 ± 0.02
	24	3.54 ± 0.02	3.37 ± 0.20	8.29 ± 0.11	N.D.	1.50 ± 0.11	1.54 ± 0.10
HPLA	0	N.D.	N.D.	N.D.	N.D.	N.D.	N.D.
	4	N.D.	N.D.	N.D.	N.D.	1.26 ± 0.14	1.21 ± 0.11
	8	4.87 ± 0.06	3.98 * ± 0.05	N.D.	N.D.	4.36 ± 0.16	4.29 * ± 0.10
	12	5.12 ± 0.05	4.69 * ± 0.02	4.77 ± 0.10	4.49 ± 0.14	7.49 ± 0.12	6.56 * ± 0.22
	24	10.23 ± 0.06	10.13 ± 0.04	5.04 ± 0.12	4.27 ± 0.04	11.73 ± 0.09	10.45 * ± 0.10
Lactic acid	0	0.09 ± 0.01	0.08 ± 0.02	0.21 ± 0.11	0.25 ± 0.01	0.05 ± 0.01	0.03 ± 0.01
	4	0.18 ± 0.05	0.21 ± 0.07	0.35 ± 0.12	0.31 ± 0.14	0.13 ± 0.02	0.15 ± 0.04
	8	0.27 ± 0.08	0.29 ± 0.04	0.48 ± 0.15	0.42 ± 0.01	0.32 ± 0.05	0.30 ± 0.02
	12	0.34 ± 0.01	0.30 ± 0.10	0.83 ± 0.05	0.85 ± 0.04	0.41 ± 0.05	0.44 ± 0.03
	24	0.88 ± 0.07	0.92 * ± 0.12	1.12 ± 0.10	1.30 ± 0.11	1.91 ± 0.16	1.90 ± 0.10
Acetic acid	0	0.19 ± 0.06	0.16 ± 0.07	0.09 ± 0.03	0.10 ± 0.02	N.D.	N.D.
	4	0.29 ± 0.05	0.27 ± 0.10	0.13 ± 0.02	0.12 ± 0.05	0.01 ± 0.01	0.01 ± 0.02
	8	0.34 ± 0.03	0.34 ± 0.12	0.22± 0.04	0.20± 0.02	0.04± 0.05	0.03± 0.04
	12	0.59 ± 0.04	0.58 ± 0.06	0.31 ± 0.01	0.35 ± 0.01	0.13 ± 0.01	0.10 ± 0.03
	24	2.56 ± 0.06	2.53 ± 0.06	0.45 ± 0.07	0.46 ± 0.02	0.27 ± 0.02	0.29 ± 0.02
Propionic acid	0	0.01 ± 0.07	N.D.	0.26 ± 0.10	0.30 ± 0.05	0.25 ± 0.06	0.22 ± 0.07
	4	0.48 ± 0.04	0.41 ± 0.10	0.30 ± 0.04	0.32 ± 0.03	0.26 ± 0.02	0.22 ± 0.08
	8	0.57 ± 0.02	0.48 ± 0.03	0.32 ± 0.10	0.36 ± 0.09	0.30 ± 0.01	0.25 ± 0.05
	12	0.85 ± 0.01	0.56 ± 0.05	0.35 ± 0.06	0.40 ± 0.07	0.35 ± 0.06	0.29 ± 0.02
	24	1.96 ± 0.03	1.01 * ± 0.07	0.41 ± 0.08	0.55 ± 0.06	0.37 ± 0.05	0.32 ± 0.04
Ethanol	0	N.D.	N.D.	N.D.	N.D.	N.D.	N.D.
	4	0.28 ± 0.02	0.29 ± 0.02	0.19 ± 0.04	0.18 ± 0.02	0.28 ± 0.05	0.23 ± 0.04
	8	0.32 ± 0.05	0.34 ± 0.03	0.20 ± 0.10	0.22 ± 0.02	0.37 ± 0.10	0.35 ± 0.12
	12	0.57 ± 0.03	0.59 ± 0.03	0.29 ± 0.02	0.25 ± 0.03	0.48 ± 0.12	0.49 ± 0.02
	24	3.82 ± 0.14	3.80 ± 0.04	0.59 ± 0.01	0.63 ± 0.04	0.57 ± 0.21	0.55 ± 0.01

The results are presented as mean values ± SDs (n = 3). A two-way ANOVA was applied to compare differences between the bacterial control species, along with the Tukey multiple comparisons method. A *p*-value is provided if *p* < 0.05, and the symbols for the Tukey interpretation are as follows: * *p* < 0.05; N.D.—not detected; HPLA—4-hidroxifenillactic acid; LP-C—*L. plantarum* (control); LR-C—*L. rhamnosus* (control); LF-C—*L. fermentum* (control); LP-Fe_3_O_4_—*L. plantarum* after fermentation with Fe_3_O_4_ NPs; LR-Fe_3_O_4_—*L. rhamnosus* after fermentation with Fe_3_O_4_ NPs; LF-Fe_3_O_4_—*L. fermentum* after fermentation with Fe_3_O_4_ NPs.

## Data Availability

The original contributions presented in this study are included in the article.

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
