# Peer review of "Bioconversion of Fe3O4 Nanoparticles by Probiotics"

_pharmaceuticals, 2025, doi:10.3390/ph18040542_

Round 1

Reviewer 1 Report

Comments and Suggestions for Authors

This is an interesting research article with adequate novelty. Some points sould be addressed.

  • In the Abstract, the abbreviations TEM, SEM, FT-IR, and XRD should be explained.
  • In the Introduction, in lines 57-58, the authors reported that "These bacteria lower 
    intestinal pH, enhancing iron solubility and promoting gut absorption." This statement needs a bi more analysis (e.g. how much lower intestinal pH?).
  • A statement should be added to the Introduction section concerning whether the probiotics (and especially those used in the present study) are stable in the acidic gastric environment.
  • Relevant references should be added in sections 2.3 and 2.5.2.
  • In section 2.6, did the authors use a normality test for their contunious variables? If yes, they should reported it in this section.
  • The resolution of Figure 5 should be improved.

Author Response

Dear Reviewer,

Thank you for bringing these important points to our attention. We appreciate the opportunity to address these concerns in our revised manuscript.

The corrections are marked in the manuscript in red.

Our revised sections, according to your comments, are listed as follows:

Comment 1:
This is an interesting research article with adequate novelty. Some points should be addressed.

Response to Comment 1:
We highly appreciate your consideration of the manuscript. Your comments and suggestions helped us improve the flow of the paper and its scientific soundness. Please find below the response to each specific comment.

Comment 2:
In the Abstract, the abbreviations TEM, SEM, FT-IR, and XRD should be explained.

Response to Comment 2:
Thank you for the observation. The full names of the abbreviations have been introduced in the Abstract upon their first mention to enhance clarity for readers unfamiliar with the terms.

Comment 3:
In the Introduction, in lines 57–58, the authors reported that "These bacteria lower intestinal pH, enhancing iron solubility and promoting gut absorption." This statement needs a bit more analysis (e.g., how much lower intestinal pH?).

Response to Comment 3:
Thank you for your comment. We agree that the statement required further elaboration. We have revised the sentence to include more detail on the degree of pH reduction and its significance for iron solubility, supported by literature:

Line 66–72:
“These bacteria lower intestinal pH, primarily through acid production, which enhances iron solubility and promotes gut absorption. One of the key excreted metabolites, p-hydroxyphenyllactic acid (HPLA), produced by L. fermentum, exhibits potent ferric-reducing activity under acidic conditions. Studies have shown that HPLA efficiently reduces ferric iron (Fe³⁺) to its more bioavailable ferrous form (Fe²⁺) at pH 3.8, significantly diminishing this activity at pH values above.”

Comment 4:
A statement should be added to the Introduction section concerning whether the probiotics (and especially those used in the present study) are stable in the acidic gastric environment.

Response to Comment 4:
Thank you for the helpful suggestion. We have revised the Introduction section to address the acid stability of the probiotic strains used in this study. Specifically, we included:

Line 51–61:
“Although Fe₃O₄ NPs offer promising advantages due to their stability and biocompatibility, their absorption in the gastrointestinal tract remains a significant challenge, requiring innovative strategies to enhance their bioavailability. One such approach involves using acid-resistant probiotics that can survive gastric transit and maintain functional viability in the gut, thereby supporting nutrient absorption. L. rhamnosus, particularly strain GG, is well known for its acid tolerance and ability to adhere to intestinal epithelial cells. These properties support its survival in the gastrointestinal environment and its efficacy in promoting intestinal health and improving the absorption of micronutrients such as iron [6, 7]. Also, L. fermentum and L. plantarum have shown excellent survival in simulated gastric juice, with viability rates reaching up to 86.7%, and 96.97%, respectively [8–10].”

Comment 5:
Relevant references should be added in Sections 2.3 and 2.5.2.

Response to Comment 5:
We have carefully reviewed the methodology sections and incorporated relevant references in Sections 2.3 and 2.5.2 to support the described procedures.

Comment 6:
In Section 2.6, did the authors use a normality test for their continuous variables? If yes, they should report it in this section.

Response to Comment 6:
We appreciate your observation. We have revised Section 2.6 (Statistical Analysis) to include the use of normality testing:

“All the values were displayed as means ± standard deviations (SD) from three independent experiments. All continuous variables were tested for normality using the Shapiro–Wilk test before statistical analysis. Data following a normal distribution were analyzed using one-way analysis of variance (ANOVA), Tukey’s comparison test, and post hoc Dunnett’s multiple comparison tests via Minitab statistical software (version 16.1.0; LEAD Technologies, Inc., Charlotte, NC, USA). Graph Prism Version 8.0.1 (GraphPad Software Inc., San Diego, CA, USA) was also used to analyze differences among samples, with significance set at p < 0.05. Statistical significance was assumed at the 95% confidence level.”

Comment 7:
The resolution of Figure 5 should be improved.

Response to Comment 7:
We have replaced Figure 5 with a higher-resolution version to ensure improved clarity and visual quality.

Reviewer 2 Report

Comments and Suggestions for Authors

The following are some suggestions for major changes to the article

1. The introduction section could be supplemented with the latest mechanisms of probiotic-mediated iron metabolism regulation (refer to the provided references).

2. In the Materials and Methods section, for nanoparticle synthesis, the ratio of ethanol to ethyl acetate in the washing steps (e.g., "4:1" – is it a volume ratio?) should be specified, and whether residual reagents were detected should be addressed. TEM/SEM sample preparation should include details on fixation and drying conditions (e.g., whether critical point drying was used) to avoid artifacts.

3. HPLC analysis should provide calibration curves and detection limits for the standards to validate the reliability of metabolite quantification. The control group setup should clarify whether a blank nanoparticle group (without bacteria) was included to exclude the influence of nanoparticle self-oxidation.

4. The background color of the TEM images in Figure 4 should be standardized. Ensure consistent font styles across all figures and tables. Functional validation of 4-HPLA should be supplemented with in vitro experiments demonstrating its direct promotion of iron absorption. The discussion on oxidation mechanisms should differentiate between chemical oxidation and microbially-mediated redox processes.

5. In the Data and Statistics section, for Table 1, unify the notation for values such as "ND" and provide detection limits. For particle size statistics, specify the number of particles analyzed (n≥100) in TEM images to ensure the reliability of Gaussian fitting.

6. In the Conclusions section, the limitations of the study (e.g., differences between in vitro experiments and in vivo environments) should be emphasized, and future validation through animal models or clinical trials is recommended.

7. It is suggested to include the latest literature (2023–2024) on probiotic-nanoparticle interactions.

Comments on the Quality of English Language

The English can be improved.

Author Response

Dear Reviewer,

Thank you for your thorough and insightful comments. We appreciate your time and the constructive suggestions that have helped improve the scientific rigor and clarity of our manuscript. Below are our detailed responses to each of your comments. All modifications have been incorporated into the revised manuscript and marked in red.

Comment 1:
The introduction section could be supplemented with the latest mechanisms of probiotic-mediated iron metabolism regulation (refer to the provided references).

Response to Comment 1:
Thank you for your valuable suggestion regarding the inclusion of recent mechanisms related to probiotic-mediated iron metabolism regulation. We fully agree that this addition enhances the depth and relevance of our manuscript. Based on the references you provided, we have supplemented the Introduction section with recent findings on molecular mechanisms:

Line 80–98:
“Recent advances in probiotic research have contributed to a deeper understanding of the molecular pathways involved in the probiotic-mediated regulation of iron metabolism [21]. Beyond increasing solubility and reducing ferric iron to its more bioavailable ferrous form, specific probiotic strains have been shown to modulate the expression of host genes involved in iron transport and storage [18, 22]. For example, the upregulation of divalent metal transporter 1 (DMT1) and ferroportin, as well as the modulation of ferritin expression, plays a crucial role in enhancing cellular iron uptake and systemic distribution [23]. Moreover, probiotics have demonstrated the ability to influence signaling pathways such as TLR2/NF-κB, which are involved in immune response and inflammatory regulation, thereby indirectly promoting a gut environment more conducive to iron absorption [3, 11, 13]. In addition, probiotics contribute to intestinal barrier integrity by modulating the expression of mucin and tight junction proteins such as MUC2, CLDN, and ZO-1, which facilitate nutrient permeability and reduce inflammation [17]. Their antioxidant properties also play a role in mitigating oxidative stress associated with iron metabolism [24]. Furthermore, emerging evidence suggests that certain lactic acid bacteria may utilize extracellular electron transfer mechanisms to enhance ferric iron reduction, representing a novel metabolic pathway that supports iron solubilization and absorption [13].”

Comment 2:
In the Materials and Methods section, for nanoparticle synthesis, the ratio of ethanol to ethyl acetate in the washing steps (e.g., "4:1" – is it a volume ratio?) should be specified, and whether residual reagents were detected should be addressed. TEM/SEM sample preparation should include details on fixation and drying conditions (e.g., whether critical point drying was used) to avoid artifacts.

Response to Comment 2:
Thank you for the insightful suggestions. We have addressed all points in the revised Materials and Methods section:

  • We clarified that the “4:1” ratio (ethanol: ethyl acetate) refers to a volume ratio (v/v).
  • We confirmed that no residual reagents were detected post-washing, as verified via FT-IR.
  • We expanded the TEM/SEM preparation protocol. The updated section reads:
    “Transmission electron microscopy (TEM) and scanning electron microscopy (SEM) analyses were conducted using a HITACHI HD-2700 STEM microscope (Hitachi, Tokyo, Japan) equipped with a digital image recording system alongside an SU8230 SEM from the same manufacturer. Dehydration was achieved using an ethanol series (30%), with SEM samples dried using a liquid CO₂ critical point drier (1,200 psi, 31 °C, 30 min) to eliminate surface-tension artifacts. To prevent aggregation, TEM samples were air-dried on carbon-coated 400-mesh copper grids under low humidity with 0.1% BSA. The microscope was coupled with an Aztec X-Max 1160 EDX detector (Oxford Instruments, Abingdon, UK) for energy-dispersive X-ray spectroscopy (EDS). SEM/EDS images were acquired under 30 kV and 10 μA operating conditions. Image acquisition and particle size measurement were performed using Hitachi acquisition software (version 8.1), and average particle size was determined using Gaussian fitting in Origin 2019b [23].”

Comment 3:
HPLC analysis should provide calibration curves and detection limits for the standards to validate the reliability of metabolite quantification. The control group setup should clarify whether a blank nanoparticle group (without bacteria) was included to exclude the influence of nanoparticle self-oxidation.

Response to Comment 3:
We sincerely appreciate this observation. We have revised the HPLC section as follows:

  • Calibration curves for each metabolite (with R² > 0.99) and detection limits are now described in detail. These are also included in the Supplementary Files for transparency and reproducibility.
  • We clarified that a blank nanoparticle group was not included in the HPLC analysis. This was based on FT-IR and XRD characterizations, which confirmed that Fe₃O₄ nanoparticles remained stable in aqueous solution and did not release oxidized byproducts. Supporting literature confirms the absence of nanoparticle self-oxidation under the tested conditions.

Comment 4:
The background color of the TEM images in Figure 4 should be standardized. Ensure consistent font styles across all figures and tables. Functional validation of 4-HPLA should be supplemented with in vitro experiments demonstrating its direct promotion of iron absorption. The discussion on oxidation mechanisms should differentiate between chemical oxidation and microbially-mediated redox processes.

Response to Comment 4:
Thank you for your comprehensive feedback. We have:

  • Standardized the background color of the TEM images in Figure 4.
  • Unified font styles across all figures and tables for consistency.
  • Acknowledged the need for functional validation of 4-HPLA and included this as a direction for future work. While in vitro validation was not performed in this study, we now reference prior studies supporting its iron-reducing capability.
  • Revised the discussion to differentiate between chemical oxidation (e.g., Fe₃O₄ oxidation under acidic conditions) and microbially mediated redox processes (e.g., enzymatic reduction by probiotics and organic acid excretion).

Comment 5:
In the Data and Statistics section, for Table 1, unify the notation for values such as "ND" and provide detection limits. For particle size statistics, specify the number of particles analyzed (n≥100) in TEM images to ensure the reliability of Gaussian fitting.

Response to Comment 5:
We appreciate your careful attention to detail. The following updates were made:

  • We standardized the notation to “N.D. – not detected” in Table 1.
  • We added a clarification in the Materials and Methods section, stating:
    Lines 130–132:
    “To ensure statistical reliability, n≥100 particles were measured across multiple images, and the data were fitted to a Gaussian distribution using Origin 2019b [23].”

Comment 6:
In the Conclusions section, the limitations of the study (e.g., differences between in vitro experiments and in vivo environments) should be emphasized, and future validation through animal models or clinical trials is recommended.

Response to Comment 6:
Thank you for this important point. We have now added a dedicated Limitations subsection before the Conclusions section:

“4. Limitations”
“This study's findings should be interpreted with consideration of its limitations. First, although 4-HPLA has been identified as a key metabolite in iron reduction, direct functional validation of its role in promoting iron absorption was not conducted. Future in vitro assays using intestinal cell models must confirm its biological activity. Second, additional in vivo studies are necessary to assess the impact of probiotic-fermented Fe₃O₄ NPs on iron bioavailability. Third, the results were obtained using specific probiotic strains (L. fermentum, L. plantarum, and L. rhamnosus), and further studies should investigate whether strain-specific metabolic variations influence their ability to enhance iron absorption.”

Comment 7:
It is suggested to include the latest literature (2023–2024) on probiotic-nanoparticle interactions.

Response to Comment 7:
Thank you for the recommendation. We have reviewed recent publications and updated the manuscript to include several new references from 2023–2024 related to probiotic–nanoparticle interactions.

Reviewer 3 Report

Comments and Suggestions for Authors

This paper noted aims to investigate the underlying molecular and biochemical mechanisms by which select probiotic strains, L. fermentum, L. rhamnosus, and L. plantarum, synergistically interact with Fe₃O₄ NPs to enhance iron solubilization. The study is relatively reasonable and innovative. However, there are still some problems with the presentation The detailed comments are listed as following:

  1. In line 205, additional broad absorption features between 3000 and 3500 cm⁻¹, however, in the figure 2A, the absorption features didn’t exist.
  2. What’s the limitation of your research? You should describe in details at last.
  3. What’s the mechanism of the synergistic effect between probiotics and iron nanoparticles?
  4. How to evaluate the improvement of iron solubility by using the probiotics?
  5. When discussing the mechanism of differences in the effects of different probiotic strains on the Fe₃O₄ nanoparticles, in addition to factors such as cell surface charge and hydrophilicity, a more comprehensive analysis can be carried out in combination with the differences in gene expression and metabolic pathway characteristics of each strain to reveal the intrinsic mechanism more deeply.
  6. In the part of “2.4. Probiotics Inoculum Preparation with Fe3O4 NPs”, the authors should describe the method in details. It’s too simple.

Author Response

Dear reviewer,

We are grateful for your valuable comments, which improve the quality of the paper. Please be aware that the corrections are marked in the manuscript in red. Our revised sections, according to your comments, are listed as follows:

Comment 1:
This paper noted aims to investigate the underlying molecular and biochemical mechanisms by which select probiotic strains, L. fermentum, L. rhamnosus, and L. plantarum, synergistically interact with Fe₃O₄ NPs to enhance iron solubilization. The study is relatively reasonable and innovative. However, there are still some problems with the presentation. The detailed comments are listed as following:

Response to Comment 1:
We highly appreciate your consideration of the manuscript. Your comments and suggestions helped us improve the flow of the paper and its scientific soundness. Please find below the response to each comment and suggestion.

Comment 2:
In line 205, additional broad absorption features between 3000 and 3500 cm⁻¹, however, in the figure 2A, the absorption features didn't exist.

Response to Comment 2:
Thank you for the observation. The information has been removed.

Comment 3:
What's the limitation of your research? You should describe in detail at last.

Response to Comment 3:
We greatly appreciate your suggestion to elaborate on our study's limitations. We have added a dedicated Limitations subsection before the Conclusions section.

Comment 4:
What's the mechanism of the synergistic effect between probiotics and iron nanoparticles?

Response to Comment 4:
Thank you for your insightful question regarding the mechanism of the synergistic effect between probiotics and Fe₃O₄ nanoparticles. We have expanded the Discussion section to provide a detailed mechanistic explanation. Relevant content added includes:

  • Line 261–265:
    "Also, probiotics have been reported to adhere to the surface of Fe₃O₄ nanoparticles, enhancing their solubility and bioavailability [39, 40]. Lactobacillus and Bifidobacterium species have been stated to possess surface structures (exopolysaccharides, lipoteichoic acids, surface proteins) that enable adhesion to mineral surfaces [40]."
  • Line 352–370:
    "Moreover, probiotic fermentation contributes to gut acidification by producing organic acids (short-chain fatty acids), which enhances iron solubility and absorption [52]... These mechanisms collectively contribute to improved iron absorption and probiotic viability, resulting in a synergistic effect."

Comment 5:
How to evaluate the improvement of iron solubility by using the probiotics?

Response to Comment 5:
Fe²⁺ is more soluble and absorbable. The measurement can be indirect by assessing how probiotics affect the Fe³⁺ → Fe²⁺ conversion.
It can also be measured by an in vitro absorption trial using a Caco-2 cell model.
Another indirect measurement relies on probiotics releasing metabolites (e.g., lactic acid, short-chain fatty acids) that can improve iron solubility by lowering pH and breaking down phytic acid.

Comment 6:
When discussing the mechanism of differences in the effects of different probiotic strains on the Fe₃O₄ nanoparticles, in addition to factors such as cell surface charge and hydrophilicity, a more comprehensive analysis can be carried out in combination with the differences in gene expression and metabolic pathway characteristics of each strain to reveal the intrinsic mechanism more deeply.

Response to Comment 6:
Thank you for your insightful suggestion. In addition to cell surface charge and hydrophilicity, we discuss potential gene expression and metabolic pathway differences. Specifically:

  • Line 297–311:
    "...Beyond these physicochemical properties, strain-specific gene expression and metabolic pathways differences may explain these effects [54, 55]... Additionally, gene expression in metal ion homeostasis may contribute to intracellular iron sequestration and storage [57, 58]."

Comment 7:
In the part of "2.4. Probiotics Inoculum Preparation with Fe₃O₄ NPs", the authors should describe the method in detail. It's too simple.

Response to Comment 7:
Thank you for your insightful comment on the need for a more detailed description of the method for preparing the Fe₃O₄ NPs probiotic inoculum. We have revised Section 2.4 to provide clearer and more detailed information:

"Lactic acid bacteria strains L. fermentum LMG 6902, L. rhamnosus LMG 25626, and L. plantarum ATCC 8014 were cultivated... All these processes were carried out in a sterile environment."

Reviewer 4 Report

Comments and Suggestions for Authors

This paper noted Bioconversion of Fe3O4 Nanoparticles by Probiotics is relatively reasonable and innovative. However, there are still some problems with the presentation of the article.

The detailed comments are listed as following:

  1. Materials and Methods: Why was a concentration of 120 µg/mL of Fe3O4 nanoparticles used for the experiment, and was a gradient tested for effects at different concentrations? If so, please add data.
  2. The figures and tables are clear and well-labeled. However, the resolution of some images (e.g., TEM) could be improved for better clarity.
  3. The current study is mainly based on in vitro experiments, and it is recommended that animal experiments or clinical trials be conducted in subsequent studies to increase the accuracy and reliability of the results.
  4. Studies have focused on the effect of probiotics on iron bioavailability, but on whether this intervention affects other functions of the probiotics themselves (e.g., immunomodulation, gut barrier function, etc.)
  5. Is the stability and activity of probiotics affected during the interaction with nanoparticles? Is there a need for further studies on the survival and functional maintenance of probiotics under different environmental conditions?
  6. Please standardize the format of references. For example, some references use“: p.” between the journal title and volume number, while others do not. Please correctly standardize the formatting of DOI links for references.
  7. Line314, Please write “N.D.- not detected” correctly .
  8. Please standardize the format of writing some data in “Table 1”. Spaces for the“±”symbol.
  9. Please use one expression consistently. For example, “Fe3O4 NPs” and “Fe3O4 nanoparticles”.
  10. Please format figure 5 correctly so that all data can be clearly presented.
  11. Please standardize the format of abbreviated names. Such as, LP-C- L. plantarum (control); LR-C- L. rhamnosus (control); LF-C for L. fermentum (control).

Author Response

Dear Reviewer,

We sincerely appreciate your valuable comments and suggestions, which have helped us improve the quality and clarity of our manuscript. The corrections have been implemented and marked in the revised document. Below, we provide detailed responses to each of your comments.

Comment 1:

This paper noted that the Bioconversion of Fe₃O₄ Nanoparticles by Probiotics is relatively reasonable and innovative. However, there are still some problems with the presentation of the article. The detailed comments are listed as follows:

Response:
We highly appreciate your recognition of the manuscript’s contribution and novelty. Your comments and suggestions have been extremely helpful in improving both the flow and scientific rigor of our study. Please find below our responses to each specific comment.

Comment 2:

Materials and Methods: Why was a concentration of 120 µg/mL of Fe₃O₄ nanoparticles used for the experiment, and was a gradient tested for effects at different concentrations? If so, please add data.

Response:
Thank you for your insightful comment. The concentration of 120 µg/mL Fe₃O₄ nanoparticles was selected based on the findings of Tafazzoli, Ghavami, and Khosravi-Darani (2024), who investigated the impact of process variables on the production of iron-enriched Saccharomyces boulardii. Their study identified this concentration as effective in facilitating iron uptake by probiotics while maintaining cellular viability. Given the relevance of their work to our experimental model, we adopted this concentration as a reference. Additionally, preliminary screening was conducted to ensure that the selected concentration did not exhibit cytotoxic effects.

Comment 3:

The figures and tables are clear and well-labeled. However, the resolution of some images (e.g., TEM) could be improved for better clarity.

Response:
We appreciate your suggestion. We have enhanced the image quality to improve clarity while ensuring that structural details remain accurately represented.

Comment 4:

The current study is mainly based on in vitro experiments, and it is recommended that animal experiments or clinical trials be conducted in subsequent studies to increase the accuracy and reliability of the results.

Response:
Thank you for your valuable suggestion. In addition to the in vitro experiments using Caco-2 cells, we have also conducted an in vivo study utilizing a rat anemic model to further investigate the effects of iron oxide nanoparticles on iron absorption.

These results are included in a separate manuscript currently under review. Given the depth and scope of these findings, we opted to present them in a dedicated publication. However, we have now clarified this aspect in our current manuscript to ensure transparency regarding the broader context of our research.

Comment 5:

Studies have focused on the effect of probiotics on iron bioavailability, but on whether this intervention affects other functions of the probiotics themselves (e.g., immunomodulation, gut barrier function, etc.).

Response:
Thank you for your insightful observation. We agree that while numerous studies have examined the impact of probiotics on iron bioavailability, fewer have addressed whether iron supplementation in combination with probiotics may influence other probiotic-associated functions, such as immunomodulation, gut barrier integrity, or host–microbiota interactions. This is an important point that adds depth to the current discussion on the multifaceted role of probiotics.

To address this, we have added a brief discussion in the revised manuscript (see the Limitations section), acknowledging this gap in the literature and suggesting that future studies should investigate potential trade-offs or synergistic effects between iron bioavailability and other probiotic-mediated functions.

Comment 6:

Please standardize the format of references. For example, some references use “: p.” between the journal title and volume number, while others do not. Please correctly standardize the formatting of DOI links for references.

Response:
We have carefully reviewed and standardized the format of all references to ensure consistency, including the proper formatting of volume numbers and DOI links.

Comment 7:

Line 314: Please write “N.D.– not detected” correctly.

Response:
We sincerely appreciate your careful attention to detail. We have corrected the notation for “N.D.– not detected” accordingly.

Comment 8:

Please standardize the format of writing some data in “Table 1”, specifically the spaces for the “±” symbol.

Response:
The formatting of Table 1 has been standardized, ensuring uniformity in spacing and the placement of the “±” symbol.

Comment 9:

Please use one expression consistently. For example, “Fe₃O₄ NPs” and “Fe₃O₄ nanoparticles”.

Response:
We appreciate your suggestion to maintain consistency in terminology. In response, we have standardized the expression throughout the manuscript, consistently using “Fe₃O₄ NPs”.

Comment 10:

Please format Figure 5 correctly so that all data can be clearly presented.

Response:
Thank you for pointing this out. We have revised Figure 5 to ensure that all data are clearly visible and properly formatted. The updated figure now includes improved axis labels, adjusted scaling, and an enhanced layout for better readability. We believe these changes improve the clarity and overall presentation of the data.

Comment 11:

Please standardize the format of abbreviated names. Such as LP-C for L. plantarum (control), LR-C for L. rhamnosus (control), and LF-C for L. fermentum (control).

Response:
We have carefully reviewed and standardized the format of all abbreviated names throughout the manuscript to ensure consistency and clarity.

Round 2

Reviewer 2 Report

Comments and Suggestions for Authors

this work can be accepted

Reviewer 3 Report

Comments and Suggestions for Authors

None, the authors have reosponded my comments well.

Reviewer 4 Report

Comments and Suggestions for Authors

The format of references are not uniform, such as the format of journal name, using abbreviations or not?